
# Determination of the Optimal Lake-Marsh Pattern in the Lake-Marsh Wetland System based on Ecological Land Use and Ecological Water Use

Wuxia Bi[1,2], Baisha Weng[1], Denghua Yan[1], Meng Li[1,3], Zhilei Yu[1,3], Lin Wang[1], and Hao Wang[1]

[1]State Key Laboratory of Simulation and Regulation of Water Cycle in River Basin, China Institute of Water Resources and Hydropower Research, Beijing, 100038, China
[2]College of Hydrology and Water Resources, Hohai University, Nanjing, 210098, China
[3]Institute of Water Resources and Hydrology, Department of Hydraulic Engineering, Tsinghua University, Beijing, 100084, China

*Correspondence to*: Baisha Weng (wengbs@iwhr.com), Denghua Yan (yandh@iwhr.com)

**Abstract.** The "land use" and "water use" competitions appear more seriously between lake-marsh wetland system and its surrounding socio-economic system, also inside the lake-marsh wetland system. However, few studies focus on solving the contradictions mentioned above. While this study proposed an optimal lake-marsh pattern determination method based on the joint regulation of ecological land and ecological water, on considering the ecological services values and water shortage amount in the lake-marsh wetland system. The optimal lake-marsh pattern determination should explore the land and water demand of the protection objects, also the water supply and water demand (especially evapotranspiration) in annual and monthly time-scale. Calculation and analysis were performed for the optimal pattern of the Wolonghu Wetlands as an example. The results mainly showed that the optimal lake-marsh pattern of the Wolonghu Wetlands can be obtained with the area ratio of lake-marsh equaling to 0.651 : 0.349 and the corresponding water level of 87.4 m in annual scale. In monthly scale, except July and August, the optimal area ratio of lake and marsh varied from 0.634 : 0.366 to 0.738 : 0.262. This study could provide references for the Wolonghu Wetlands management, also for similar lake-marsh wetland system and other ecological system.

## 1 Introduction

Lake-marsh wetland system, containing lake waters and marsh around the shore of lakes or shallow lakes, as the one of the most important ecosystems on Earth, are characterized by wide distribution and large area (William and James, 2015; Xu et al., 2018). Both lake and marsh play important roles in the lake-marsh wetland system, with multiple important ecological service functions. Lake in the system can provide provisioning services (e.g., fresh water provision, fishery products provision), regulating services (e.g., water purification, climatic regulation), and supporting services (e.g., habitat for wildlife) (Boyd and Banzhaf, 2007; de Groot et al., 2012). For marsh wetland in the system, it provides provisioning services (e.g., plant products provision), regulating services (e.g., pollution degradation), supporting services (e.g., habitat for wildlife, biodiversity protection), and cultural services (e.g., landscape aesthetics) (Nahlik et al., 2012; Xu et al., 2018).



The degradation of lake-marsh wetland system has become more and more serious (Cyranoski, 2009; Davidson, 2014; Ellis and Ramankutty, 2008; Meng et al., 2017; Qiu, 2011; Stirling, 2011) due to the ecological water deficits and unreasonable development, which results in a series of ecological and environmental problems, such as, ecological functions degradation, trend of single structure, strong vulnerability, etc. (Wang et al., 2006; Zhao and Gao, 2007; Zheng et al., 2012). The lake-

marsh wetland system degradation also can lead to frequent occurrences of global warming, reduced biodiversity, and floods disasters (Costanza, 2006; Engelhardt and Ritchie, 2002; Jiang et al., 2017; Woodward and Wui, 2001). The increasing population and rapid developing economy cause the contradiction between human and lake-marsh wetland system for water and land. The global climate change would further intensify the contraction. As more and more attention paid to ecological environmental protection, several policy measures have also been proposed to protect and restore the lake-marsh wetlands.

The ideal state is to recover the lake-marsh wetlands into the original pattern in the history. However, taking into account the irresistible impact of human activities and economic development demand, it may bring out counterproductive effects if we resist on restoring the lake-marsh wetlands to historical steady state in actual. The lake-marsh wetland restoration practice should consider the conditions of both ecological system health and economic development.

Under natural conditions, lake and surrounding marsh wetland always have close hydrological and ecological links in the lake-

marsh wetland system. While the horizontal, lengthwise and vertical connectivity between lake and marsh wetland has undergone dramatic changes under the influence of high-intensity human activities, especially the rapid socio-economy development on the land resources demand and massive water use. Thus, the key point to determine the optimal lake-marsh pattern is to solve the land and water competitions. Existing studies mainly focused on the water demand amount (Wei, 2016; Yang, 2002) or the wetland area demand (Kusler, 1994; Yong, 1996; Zhang, 2006) when restore the lake-marsh wetland system.

These measures only consider parts of the system functions, which cannot solve the land and water resources competition inside the lake-marsh wetland system and with the surrounding socio-economic system.

Therefore, this study concentrated on solving the above-mentioned main contradictions in the process of protecting and restoring the lake-marsh wetland system (Fig. 1). There are two levels of contradictions: first level is the "land use" and "water use" contradictions between the lake-marsh wetland system and the surrounding socio-economic development. For the "land

use" contradiction, in most cases, it is difficult to expand the overall scale of the lake-marsh wetland system. The "water use" contradiction refers to the ecological water shortage of the lake-marsh wetland system and the water demand of the surrounding socio-economic development, this competition is evident especially in the water-deficient areas. Studies shows that the urban development has become a primary cause of the wetland loss (Iniesta-Arandia et al., 2014; Rojas et al., 2019). Zhang et al. revealed that the size of the Baiyangdian Wetland decreased during 1984 to 2013 mainly due the increased land and water

demand for local social and economic developments (Zhang et al., 2016). Second level is the "land use" contradiction in the lake-marsh wetland system, between lake and marsh wetland (Wu et al., 2012). Under the premise of the limited lake-marsh wetland system scale and the total water use amount, how to meet the water demand of the lake-marsh wetland system as much as possible and to play its ecological service function is the key scientific question to be solved. One of the effective ways to solve the above problem is determining the optimal area pattern of lake and marsh in the lake-marsh wetland system. Our





previous study recommended accelerating ecological land-use planning and strengthening the regulation of ecological water use on river system (Yan et al., 2012). This study focuses on the ecological land use management and ecological water use regulation of the lake-marsh wetland system.

The main objectives of our study were to: i) estimate the water demand and supply amount of the lake-marsh wetland system; ii) evaluate the ecological services values of the lake-marsh wetland system; iii) propose a method to determinate the optimal lake-marsh pattern based on the balance of ecological services values and water shortage amount; iv) apply the above theories to the optimal lake-marsh pattern determination in the Wonlonghu Wetlands. Different from previous studies mainly concerned on water amount (Cui et al., 2009; Yang and Yang, 2010; Yang and Chen, 2011), this method concerned the maximum ecological services benefits and minimum water shortage amount under the premise of limited area of lake-marsh wetland system and the total water use amount. The broad implication of the present research is to better establish the lake-marsh pattern in the Wolonghu Wetlands. The results could help to deliver some recommendations on ecological construction policy programs in local wetlands for relevant departments and researchers, also would provide references for the specific pattern implement of local and other similar ecological systems.

## 2 Methodology

### 2.1 Determination of Optimal Lake-Marsh Pattern

For a certain lake-marsh wetland system, to some extent, the area ration of lake and marsh is fixed based on the lake bottom elevation. To determine the optimal lake-marsh pattern, the core is minimizing the competitive land and water use with meeting the space and water level demand of protection objects in lake-marsh wetland system, which usually needs some projects to get the optimal pattern. Meanwhile, the determination of optimal lake-marsh pattern needs some constraints which can reflect the characteristics and ecological function of lake-marsh wetland system. To estimate the ecological functions of ecosystem, the ecological services values are usually evaluated (Pechacek et al., 2013). With the fixed area of lake and marsh, the ecological services values of lake-marsh wetland system are larger with large marsh area, while the ecological water demand cannot be met with small lake area (low water storage capacity). Meanwhile, the evapotranspiration augments with large lake area, especially in plain area where the water consumption is large. Therefore, in this study, the optimal lake-marsh pattern was determined by the possible minimum water shortage amount and the possible maximum ecological services values with meeting the required water levels of different protection objects.

The specific calculation process can be found in Fig. 2, which includes six parts: i) determine the protection objects; ii) analyze the required water level of each protection object; iii) determine min L and max L; iv) calculate unit ecological services values of lake and marsh; v) calculate the water demand amount and water supply amount; vi) obtain the optimal area ratio of lake and marsh, also the corresponding water level. Notably, the total water surface area of lake and marsh is definite for a certain water level. As the water level and protection objects change in different time scales and time periods, this paper explored the





optimal lake-marsh pattern based on the balance of maximum ecological services values and minimum water shortage amount in annual and monthly scales. Eq. (1)-(12) show the objective functions and constraints.

The objective functions can be expressed as Eq. (1) and (2):

$$\max V = V_{\text{lake}} + V_{\text{marsh}} \tag{1}$$

$$\min Q = Q_{\text{D}} - Q_{\text{S}} \tag{2}$$

Where $V$ is the ecological services values of lake-marsh wetland system, $V_{\text{lake}}$ is the ecological services values of lake, $V_{\text{marsh}}$ is the ecological services values of marsh, $Q$ is the water shortage amount, $Q_{\text{D}}$ is the water demand amount, $Q_{\text{S}}$ is the water supply amount.

For the constraints, we first defined the water volume corresponding to the lake-marsh pattern (Eq. (3) and (4)):

$$V_{\text{lake}} = P_{\text{lake}} Q_{\text{lake}} = P_{\text{lake}} f(A_{\text{lake}}) \tag{3}$$

$$V_{\text{marsh}} = P_{\text{marsh}} A_{\text{marsh}} \tag{4}$$

Where $V_{\text{lake}}$ is the ecological services values of lake, $V_{\text{marsh}}$ is the ecological services values of marsh, $P_{\text{lake}}$ is the unit ecological service value of lake, $Q_{\text{lake}}$ is the water amount in the lake, $A_{\text{lake}}$ is the area of lake, $P_{\text{marsh}}$ is the unit ecological service value of marsh, $A_{\text{marsh}}$ is the area of marsh.

Then we could obtain the lake-marsh pattern by the area ratio of lake and marsh (Eq. (5)):

$$A = A_{\text{lake}} + A_{\text{marsh}} = \alpha_{\text{lake}} A + \alpha_{\text{marsh}} A \,, \begin{cases} A \leq A_{\text{max}} \\ \alpha_{\text{lake}} + \alpha_{\text{marsh}} = 1 \\ 0 < \alpha_{\text{lake}}, \alpha_{\text{marsh}} < 1 \end{cases} \tag{5}$$

Where $A$ is the total area of lake and marsh, $A_{\text{lake}}$ is the area of lake, $A_{\text{marsh}}$ is the area of marsh, $\alpha_{\text{lake}}$ is the ratio of lake area, $\alpha_{marsh}$ is the ratio of marsh area, $A_{\text{max}}$ is the maximum area of the lake-marsh wetland system.

Next, to calculate the water shortage amount, firstly we should obtain the water demand amount (Eq. (6)-(8)):

$$Q_{\text{D}} = D_{\text{lake}} + D_{\text{marsh}} + V_{\text{t}} \tag{6}$$

$$D_{\text{lake}} = E_{\text{lake}} + I_{\text{lake}} = \alpha A_{\text{lake}} + I_{\text{lake}} \tag{7}$$

$$D_{\text{marsh}} = E_{\text{plant}} + E_{\text{marsh}} + I_{\text{marsh}} = \beta A_{\text{plant}} + \gamma A_{\text{marsh}} + I_{\text{marsh}} = \delta A_{\text{marsh}} + I_{\text{marsh}} \tag{8}$$

Where $Q_{\text{D}}$ is the water demand amount, $D_{\text{lake}}$ is the water demand amount of lake, $V_{\text{t}}$ is the water volume of the lake-marsh wetland system, $E_{\text{lake}}$ is the water evapotranspiration of lake, $I_{\text{lake}}$ is the infiltration of lake, $\alpha$ is the evapotranspiration coefficient of lake, $D_{\text{marsh}}$ is the water demand amount of marsh, $E_{\text{plant}}$ is the plant evapotranspiration coefficient of marsh, $E_{\text{marsh}}$ is the water evapotranspiration coefficient of marsh, $I_{\text{marsh}}$ is the infiltration of marsh, $\beta$ is the plant evapotranspiration





coefficient of marsh, $\gamma$ the water evapotranspiration coefficient of marsh, $\delta$ is the comprehensive evapotranspiration coefficient of marsh.

The water supply amount can be calculated by Eq. (9). The water shortage amount is the difference of the water demand amount and the water supply amount.

$$Q_S = P + R_S + R_G + V_e \qquad (9)$$

Where $Q_S$ is the water supply amount, $P$ is the precipitation, $R_S$ is the water supplied by surface water (not including the upstream interception), $R_G$ is the water supplied by groundwater, $V_e$ is the average existing water volume of the system.

The area and volume can be expressed by the function of the water level, and the water level is limited by the protection objects in the lake-marsh system (Eq. (10)-(12)).

$$A = f(L) \qquad (10)$$

$$V_t = f(L) \qquad (11)$$

$$L(t) = f[O_1(t), O_2(t), \dots, O_i(t), \dots], L(t) \in [\min(L), \max(L)] \qquad (12)$$

Where $A$ is the total area of lake and marsh, $V_t$ is the water volume of the lake-marsh wetland system, $L$ is the water level, $L(t)$ is the water level in different time period $t$, $O_i(t)$ is the required water level of protection object $i$ in different time period $t$, $\min(L)$ is the lowest water level demanded by the protection objects, $\max(L)$ is the highest water level demanded by the protection objects.

Notably, $A_{max}$ is determined by the local area limit, some by the limit of lakeside road, some by the limit of lake-marsh wetland area, etc. The protection objects $O_i(t)$ would change in different time periods and in different study area. To meet as far as possible the two objective functions, the optimal area ratio of lake and marsh and water level may need to be calculated multiple times. The calculation was compiled in GAMS language and solved by GAMS software package.

## 2.2 Ecological services values evaluation

The main ecological functions of lake and marsh wetlands are runoff regulation, water supply, water purification, water conservation, climate regulation, biodiversity protection and other existing values (Chen and Zhang, 2000; Ma et al., 2009; Tiner, 2010). According to the classification system of the Millennium Ecosystem Assessment (MA), the ecological services of lake-marsh wetland system include supply, regulation, support and cultural services (Groot et al., 2002; MA, 2005; Zhang et al., 2015; Zhang and Liu, 2011). The ecological services of lake contain fishery products, pollutants carrying capacity and water conservation; the ecological services of marsh include culture, plant products, carbon sequestration, sand fixation, pollution degradation and biodiversity protection. Table 1 shows the specific calculation formula of each service value.



## 3 Case study

### 3.1 Study site

The Wolonghu Wetlands (42°31′–43°02′N, 122°45′–123°37′E) is located in Kangping County, Liaoning Province, northeastern part of China (Fig. 3). The region experiences temperate continental monsoon climate. The inter-annual precipitation is unevenly distributed, and the annual average precipitation was 556.37 mm during 1961 to 2016. The precipitation was especially concentrated from June to September, accounting for 74.56% of the annual precipitation. There are mainly two rivers running into the Wonglonghu Lake, the East Malian River and the West Malian River. The Wonglonghu Lake flows into the Bajiazi River, which finally runs into the Liaohe River.

The Wolonghu Wetlands is an important ecological barrier to resist southward expansion of the Horqin Desert, also is a significant stopover and energy supplement site for the global birds migrating in the route of East Asia to Australia. Among the existing 15 species of *Gruidae* in the world, 6 of which have migrated through the Wolonghu Wetlands. Previous field survey shows that the single-day monitoring number of *Grus leucogeranus* in Wolonghu Wetlands was 2900 in 2009, accounting for 70 to 80% of the migratory *Grus leucogeranus* in the world. Therefore, the Wolonghu Wetlands has significant ecological status in Northeastern China. While the development of Kangping County (one of the poor counties in China) aggravates the land and water competition between the Wolonghu Lake and its surrounding environment, the total lake-marsh area of the Wolonghu Wetlands is limited. Meanwhile, the water interception is serious in the upstream irrigation area. Land and water competition between lake and marsh also exist inside the Wolonghu Wetlands as to protect the wetland habitat of *Grus leucogeranus*. The Wolonghu Wetlands meets the ecological water and land shortage. To protect the species diversity in the Wolonghu Wetlands, it is of great importance to study the optimal lake-marsh pattern of the Wolonghu Wetlands, which can guarantee the sustainable development of Wolonghu Wetlands.

### 3.2 Data sources

The data used in this study includes basic information of the Wolonghu Wetlands (including geographic data, area data, etc.), meteorological data (including the precipitation and temperature data from 1961 to 2016), land use data, water resources data (including water resources amount, runoff process, etc.), water quality data (including chemical oxygen demand (COD), ammonia nitrogen ($NH_3$-N) and total phosphorus (TP), etc.), planting and agricultural structure data, and other information of Wolonghu Wetlands. The specific information of the data is shown in Table 2.

## 4 Results

### 4.1 Protection objects and corresponding ecological water level demand in the Wolonghu Wetlands

The main protection objects of Wolonghu Wetlands contain wild rare and endangered birds, summer migratory birds, local fish, and wetland ecosystem itself.





The key protection objectives of fish are to protect local indigenous fish and dominant species as there are no key protected and endangered fish in the Wolonghu Wetlands. According to the minimum living space requirement method, the lowest ecological water level refers to the minimum water required for fish survival plus the lake bottom elevation. The fish culture method in the Wolonghu Wetlands is mainly the natural fish farming, with required water depth at least 1 m. Therefore, the

lowest ecological water level of the Wolonghu Lake is 87.0 m with the lake bottom elevation of 86.0 m. This water level only takes into account the needs of local fish but not considers the habitat needs of rare birds in the ecosystem. The min L should be greater than 87.0 m.

Former study reveals that to maintain the ecological function of the Wolonghu Wetlands, the minimum ecological water level is about 86.3 to 87.2 m for the Wolonghu Wetlands (Wang et al., 2008). In this study, the min L is defined as 87.2 m.

According to the species composition and quantity analysis of birds in the Wolonghu Wetlands, there are five national first-class protected species, including *Ciconia boyciana*, *Grus japonensis*, *Grus leucogeranus*, *Grus monacha* and *Ciconia nigra*. *Grus leucogeranus* is the key specie of endangered bird in the Wolonghu Wetlands, its main food is *Scirpus planiculmis Fr. Schmidt*. The lowest water level is the water level that guarantees the habitat area for the *Grus leucogeranus*. The highest water level is the water level not submerging the *Scirpus planiculmis Fr. Schmidt*. During the spring migration period of *Grus*

*leucogeranus* from March to April, most of *Scirpus planiculmis Fr. Schmidt* has just started to sprout and grow, the plant height does not exceed 40 cm. During the winter migration period from October to November, the height of *Scirpus planiculmis Fr. Schmidt* does not exceed 50 cm. Therefore, for *Grus leucogeranuss*, from March to April, the min L is 87.2 m, the max L is 87.6 m, and the marsh area should be larger than 11.42 km$^2$ (Xiao et al., 2014). Thus, the min L is 87.2 m, and the max L is 87.7 m. Considering the mean density of Grus leucogeranuss is about 0.4 to 1.25 per hm$^2$ , as there has been maximum 2854

*Grus leucogeranus* appeared in the Wolonghu Wetlands, the minimum marsh area can be calculated as about 11.42 km$^2$ with *Grus leucogeranuss* density of 0.4 per hm$^2$.

The summer migratory birds arrive successively in the Wolonghu Wetlands from the end of March to the beginning of April, then inhabit and breed in the Wolonghu Wetlands. Whether poultry or wader, they need a shallow water wetland or a water grass environment to habitat and breed. The protection of summer migratory birds should focus on meeting the critical

ecological and hydrological conditions of the breeding habitats in the breeding season from May to July. On the one hand, the water level meets the growth of the emergent plants and water plants in the surrounding shallow waters, which facilitates the nesting. On the other hand, the water level cannot be too high to submerge the water plants on the shore. The suitable water depth for emergent plants should not exceed 60 cm, which means the max L should be smaller than 87.8 m.

For the Wolonghu Lake, the water surface area (A) should be smaller than the area surrounded by the lakeside road, of 62.76

km$^2$. The corresponding water level is about 88.20 m. Thus, the max L is 88.20 m. Considering all the constraints mentioned above, the min L and max L can be defined monthly. From March to April, the min L is 87.2 m, the max L is 87.6 m. From May to July, the min L is 87.2 m, the max L is 87.8 m. From October to November, the min L and max L is 87.2 m and 87.7 m, respectively. In the other months, i.e., January, February, August, September and December, the min L is 87.2 m, the max L should not be larger than 88.2 m (Table 3).





## 4.2 Unit ecological services values of the Wolonghu Wetlands

Table 4 shows the unit ecological services values of the Wolonghu Wetlands, with lake value and marsh value of 2.69 yuan $m^{-3}$ $a^{-1}$ and 128,386,864 yuan $m^{-3}$ $a^{-1}$, respectively. Also, the specific calculation details of each formula presented in Table 1 can be found in Table 4. Remarkably, the value of $P_c$, $P_p$ and $P_b$ was determined according to the research results of Costanza

(Costanza et al., 1997, 2014). The main plant products in the Wolonghu Wetlands are *Nelumbo SP.*, *Phragmites australis* and *Potentilla anserina L.*, with biomass density of 1142.69 t $km^{-2}$ $a^{-1}$, 33,844.34 t $km^{-2}$ $a^{-1}$ and 23,231.13 t $km^{-2}$ $a^{-1}$, respectively. About 50% can be provided to the market every year, thus the value of $Y_{p,i}$ is half of the biomass density of the main plant products mentioned above. To calculate the carbon sequestration value, the $Y_{CO_2}$ and $Y_{O_2}$ were estimated based on the photosynthesis of plants. The 1 g of plant matter produced by the ecological system can fix 1.63 g $CO_2$ and release 1.2 g $O_2$.

The total plant biomass density was 58,218.16 t $km^{-2}$ $a^{-1}$, thus the $Y_{CO_2}$ and $Y_{O_2}$ was 94,895.6 t $km^{-2}$ $a^{-1}$ and 69,861.8 t $km^{-2}$ $a^{-1}$, respectively. The other values were based on the local survey data and previous study (Zheng and Wu, 2015).

## 4.3 Water demand and water supply in the Wolonghu Wetlands

In this study, we calculated the minimum average annual or monthly water shortage amount. According to Eq. (6), water demand amount contains three parts: water demand of lake, water demand of marsh, and the actual existing water volume of

the lake-marsh wetland system. In this study, as the water permeability of lake bottom is poor, the infiltration of lake $I_{lake}$ and the infiltration of marsh $I_{marsh}$ were considered as zero. Eq. (9) shows that the water supply amount includes four parts: precipitation, surface water, ground water and actual existing water amount in the lake.

In annual scale, the average annual evapotranspiration of the water surface is 1288 mm in the Wolonghu Lake, thus $\alpha$ and $\gamma$ equal to 1.288 m. The average annual evapotranspiration of *Nelumbo SP.*, *Phragmites australis* and *Potentilla anserina L.* is

1010 mm, 922 mm and 616.7 mm. Based on the actual area ratio of these plants, the transformation of plant area into the marsh area can obtain the $\beta$, of 1025.9 mm, i.e., 1.0259 m. Thus, $\delta$ equals to 2.3139 m. The water volume can be calculated by the water level (Fig. 4). Based on the metrological data from 1962 to 2017, the average annual precipitation of the Wolonghu Lake is 27,211,400 $m^3$. The average annual surface water supply amount (not including the upstream interception) is about 14,807,800 $m^3$. The average annual ground water amount is 2,945,300 $m^3$. The actual existing water volume of the Wolonghu

Lake is about 51,500,000 $m^3$. The analysis of water balance in the Wolonghu Lake can be found in the reference (Yan et al., 2009).

At monthly scale, the coefficient value of water demand and water supply varies in different months in the Wolonghu Wetlands. Table 5 presents the details of each coefficient value in each month. The meaning of the symbols in the table is the same to the equations mentioned above. Notably, $R_T$ refers to the total surface runoff, $R_I$ refers to the upstream interception, $R_S$ is the

actual water amount supplied by surface runoff, which is the difference of $R_T$ and $R_I$. $R_I$ is limited by $R_T$ in each month, if the upstream demand amount is larger than $R_T$, $R_I$ equals to $R_T$.



### 4.4 Determination of the optimal lake-marsh pattern in the Wolonghu Wetlands

To determine the optimal lake-marsh pattern in the Wolonghu Wetlands, we should first establish the correlation function between water surface area and water level (A-L curve) and the correlation function between water volume and water level (V-L curve) (Fig. 4).

As the total water surface area of lake and marsh is fixed in the lake-marsh wetland system, to determinate the optimal lake-marsh pattern, the area ratio of lake and marsh should be determined, also the corresponding water level can be obtained. Fig. 5 presents the annual optimal lake-marsh pattern of the Wolonghu Wetlands. The corresponding area ratio of lake and marsh is 0.651 : 0.349, with water level of 87.4 m.

In monthly scale, it is assumed that the monthly unit ecological services values of lake and marsh is one twelfth of the annual
unit ecological services values, thus the monthly ecological services values of lake and marsh is about 0.2242 yuan m$^{-3}$ month$^{-1}$ and 10,698,905.3 yuan km$^{-2}$ month$^{-1}$, respectively. As the coefficient of evapotranspiration of water surface and plant changes in different month, the water shortage amount of lake-marsh wetland system ($Q$) has non-fixed mathematical expression.

Based on the above calculation equations, with the limit of $A_{\text{marsh}} \geq 11.42$ km$^2$, the range of $\min(L)$ and $\max(L)$ in different month, we calculated all the possible combinations of max $V$ and min $Q$, then selected the optimal area ratio of lake and marsh
in different time period. The results are plotted in Table 6, with detailed values of each concerned parameter. It can be learned than except July and August, the optimal area ratio of lake and marsh varied from 0.634 : 0.366 to 0.738 : 0.262, the corresponding water level is between 87.43 m and 87.61 m. While in July and August, the optimal area ratio of lake and marsh is 0.783 : 0.217 and 0.855 : 0.145, with water level of 87.8 m and 88.2 m, respectively. Fig. 6 plots the monthly optimal lake-marsh pattern of the Wolonghu Wetlands. It can be learned that in the months with abundant water supply amount such as in
August, the marsh plants would be submerged; while in the months with small water supply amount, such as April, the water surface area of lake is relatively small and the marsh area will expand. It is difficult to maintain the proposed optimal lake-marsh pattern of the Wolonghu Wetlands based on the status quo without any regulation projects.

## 5 Discussion

### 5.1 The optimal lake-marsh pattern in different time scales

The method proposed in the paper is applicable for the lake-marsh wetland system with specific protection objects and water shortage problem. For the Wolonghu Wetlands, in annual scale, the optimal area ratio of lake and marsh is 0.651 : 0.349, with water level of 87.4 m. According to the reference (Jiang, 2015), the historical evolution analysis reveals it is ideal to restore the Wolonghu Wetlands to the 1980 s. The corresponding water level is 87.5 m, our result is approach to this value.

As there are scant studies on the monthly optimal lake-marsh pattern, this paper focused on the monthly process. Comparing
the monthly values with the annual value, it can be learned from Table 6 that both the value of each month and the average value of all the months are not the same to the annual value. The reason could be explained by the water supply changes in





different months, in some months with litte surface runoff, the upstream interception demand may occupy all the surface runoff, thus the surface water apply is zero in monthly scale. While in annual scale, the annual surface runoff can meet the annual upstream interception demand, thus the annual upstream interception would be larger the sum of monthly upstream interception with different calculation ways. Meanwhile, the protection objects and plants growth vary in different months, the

corresponding water demand (water level, evatranspiration and so on) will change as well. It is indicated that when design the optimal lake-marsh pattern, it is better to explore the optimal pattern in different time scales, then regulate the area ratio of lake and marsh to approprite status.

## 5.2 Suggestions for maintaining the optimal pattern

In the Wolonghu Wetlands, the water level has been decreasing in recent years (Li, 2009), it is difficult to maintain the optimal
water level analyzed above. To guarantee the wetland habitat for the endangered animals in the Wolonghu Wetland, some regulation projects can be implemented to meet the spatial management and water use control. The core issue is to maintain the optimal area ratio of lake and marsh as possible. In the months with too much water supply such as July and August, the lake can play the water storage function with the projects storing the abundant water, the marsh area can be guaranteed as well. In the months with not enough water supply, the lake can supply water to maintain the marsh area for the needing and breeding
of *Grus leucogeranus* and summer migratory birds. The specific design should consider not only the ecological and hydrology conditions, but also the economy, society and other factors. Further studies could focus on the project design to march the optimal lake-marsh pattern, especially the projects considering the harmony of nature and human, such as invisible slope.

In addition, the water demand amount was determined by the actual plants pattern of the Wolonghu Wetlands. As the evapotranspiration coefficient is different for each plant, to maintain the optimal ecological function of the system, study on
the optimal plants pattern can be effectuated, which could guide the future plants allocation in the Wolonghu Wetlands and other similar lake-marsh wetland systems.

## 5.3 Guides for ecological restoration

Facing to the "land use" and "water use" contradictions, in the process of ecological restoration, some projects may keep large lake area in order to protect the ecological water demand, while damage the ecological functions to some extent; others might
consider marsh wetlands only, which cannot meet the ecological water demand. For example, the "Marshland" in the east of the Lake Neusiedl covering the Austrian-Hungarian transboundary region has been affected most by the regional development, the area regulation has been studied in the process of ecological restoration (Hainz-Renetzeder et al., 2015). Wetlands land occupying pression can also be found in Bangalore (Ramachandra, 2001). Meanwhile, the larger lake size not present more biodiversity. Studies show surprising results that small lakes appear to have a higher chance to be in a vegetated clear state
(Scheffer and Van Nes, 2007). Thus, large lake area does not mean the better ecological status. At present, in China, it is believed that the occupation of ecological water and the destruction of wetlands land are still exist (Shen, 2001; Yu et al., 2005). Most present ecological restoration projects of lakes overemphasized on guaranteeing the ecological water demand,



which lead to the massive disappearance of lakeside zone. National-level wetlands with bird protection targets do not consider the fish structure in the lakes, lakes with protection object of fish do not emphasize the integrity of lakeside wetlands (Wang et al., 2014).

This paper attempts to better construct and manage the lake-marsh wetland system on considering the "land use" and "water use" contradictions. The contradictions are not only problems existing in the lake-marsh wetland system, but also in other aspects of ecological restoration. To better manage the lake-marsh wetland system, both the spatial ecological management and ecological water use control are needed. This study could provide references for the ecological restoration in other aspects.

## 6 Conclusion

An optimal lake-marsh pattern determination method considering the ecological services values and water shortage amount was proposed in this paper. To obtain the optimal lake-marsh pattern, the core issue is to keep the balance of the maximum ecological services values and minimum water shortage amount. This paper applied the Wolonghu Wetlands to explore the optimal lake-marsh pattern (area ratio of lake and marsh) with calculating the ecological services values, water demand amount and water supply amount of the lake-marsh wetland system. The main results showed that:

(1) The main protection objects of Wolonghu Wetlands contain wild rare and endangered birds (mainly the *Grus leucogeranuss* and its food *Scirpus planiculmis Fr. Schmidt*), summer migratory birds, local fish, and wetland ecosystem itself.

(2) As the water demand and water level demand of protection objects vary differently in annual and monthly scales, also the water supply varies in different months, we explored the optimal lake-marsh pattern in both annual scale and monthly scale. The annual optimal lake-marsh pattern can be obtained with the area ratio of lake and marsh equaling to 0.651 : 0.349 and the corresponding water level of 87.4 m. In monthly scale, except July and August, the optimal area ratio of lake and marsh varied from 0.634 : 0.366 to 0.738 : 0.262, the corresponding water level varies between 87.43 m and 87.61 m.

To better protect the ecology of the lake-marsh wetland system, a scientific and systematic optimal pattern determination method is essential. This paper proposed a method considering the ecological services values and water shortage amount, which can meet the spatial management and water use control. This study could provide guides for lake-marsh wetland system management departments, also can deliver some references for project design for maintaining the optimal pattern. However, there are probably some deficiencies in the current calculation method. To improve the method, a method considering more comprehensive affecting factors needs to be further explored.

*Data availability.* Data for observed rainfall are available upon request from China Meteorological Data Service Center (http://data.cma.cn/, last access: 21 February 2019).



*Author contributions.* Wuxia Bi drafted the manuscript. Baisha Weng and Denghua Yan designed the project and provided overall guidance. Wuxia Bi and Meng Li provided the calculation method. Wuxia Bi, Zhilei Yu and Lin Wang collected the data. Hao Wang checked the errors. Wuxia Bi and Baisha Weng finalized the manuscript. All authors reviewed the manuscript.

*Competing interests.* The authors declare that they have no conflict of interest.

*Acknowledgements.* This work was supported by the National Science Fund for Distinguished Young Scholars (No. 51725905), the National Natural Science Foundation of China (No. 91547209), the National Key Research and Development Project (No. 2017YFA0605004 and No. 2016YFA0601503). Many thanks to the relevant departments of Kangping County, Liaoning Province, China for their full support and help during this study.

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



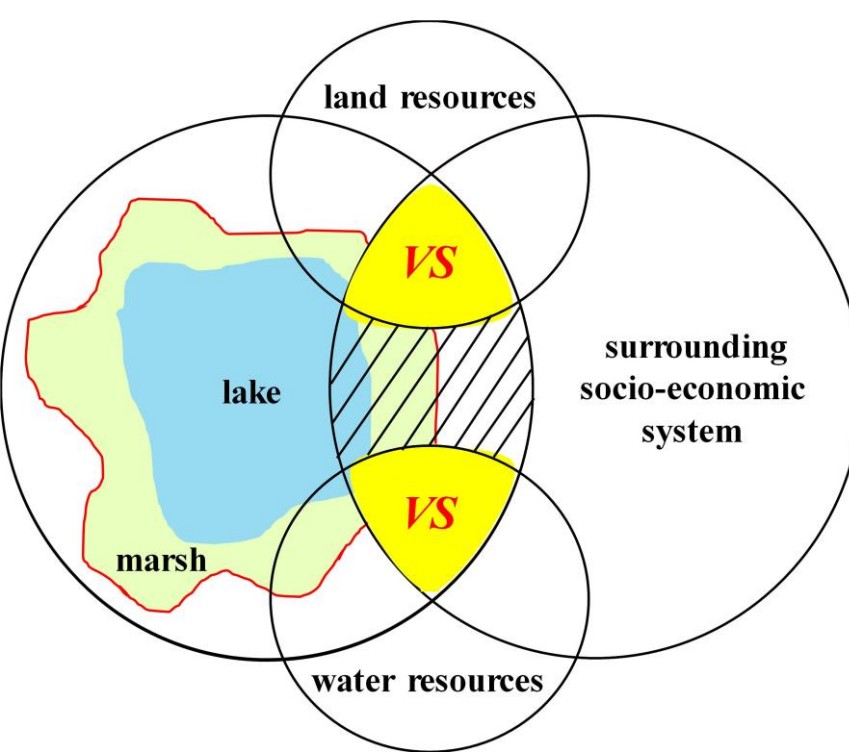

**Figure 1.** The main contradictions in the process of protecting and restoring the lake-marsh wetland system. There are mainly two contradictions between lake-marsh wetland system and its surrounding socio-economic system: land resources competition and water resources competition. Meanwhile, the land use competition also exists inside the lake-marsh wetland system.





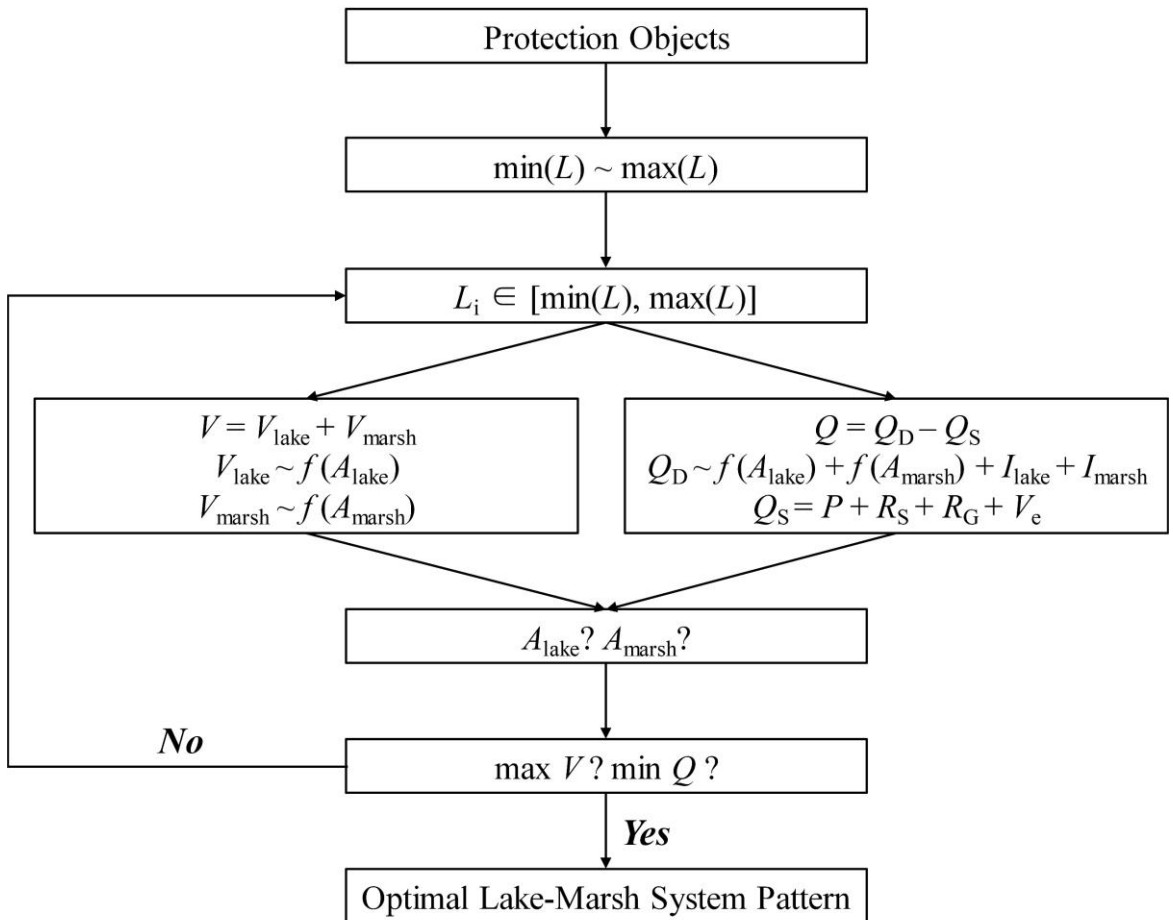

**Figure 2.** The calculation method to determine the optimal lake-marsh pattern. The $\min(L)$ and $\max(L)$ is minimum and maximum required water level of the protection objects. $L_i$ is a certain water level within the range of $\min(L)$ and $\max(L)$. $V$ refers to the ecological services values of the lake-marsh wetland system, which contains the ecological services values of lake ($V_{lake}$) and the ecological services values of marsh ($V_{marsh}$). $V_{lake}$ and $V_{marsh}$ is function of lake area ($A_{lake}$) and marsh area ($A_{marsh}$), respectively. $Q$ refers to the water shortage amount of the lake-marsh wetland system, which is the difference between the water demand amount ($Q_D$) and the water supply amount ($Q_S$). $Q_D$ is a function of $A_{lake}$, $A_{marsh}$, infiltration of lake ($I_{lake}$) and marsh ($I_{marsh}$). The $I_{lake}$ and $I_{marsh}$ can be ignored in the soils with low infiltration capacity or geological conditions with poor water permeability. $Q_S$ is a function of precipitation ($P$), surface water supply (not including the upstream interception) ($R_S$), groundwater supply ($R_G$), and the average existing water volume of the system ($V_e$). Within the range of demanded water levels, the optimal lake-marsh pattern can be obtained with the maximum ecological services values (max $V$) and minimum water shortage amount (min $Q$), which are determined by the ratio of $A_{lake}$ and $A_{marsh}$.

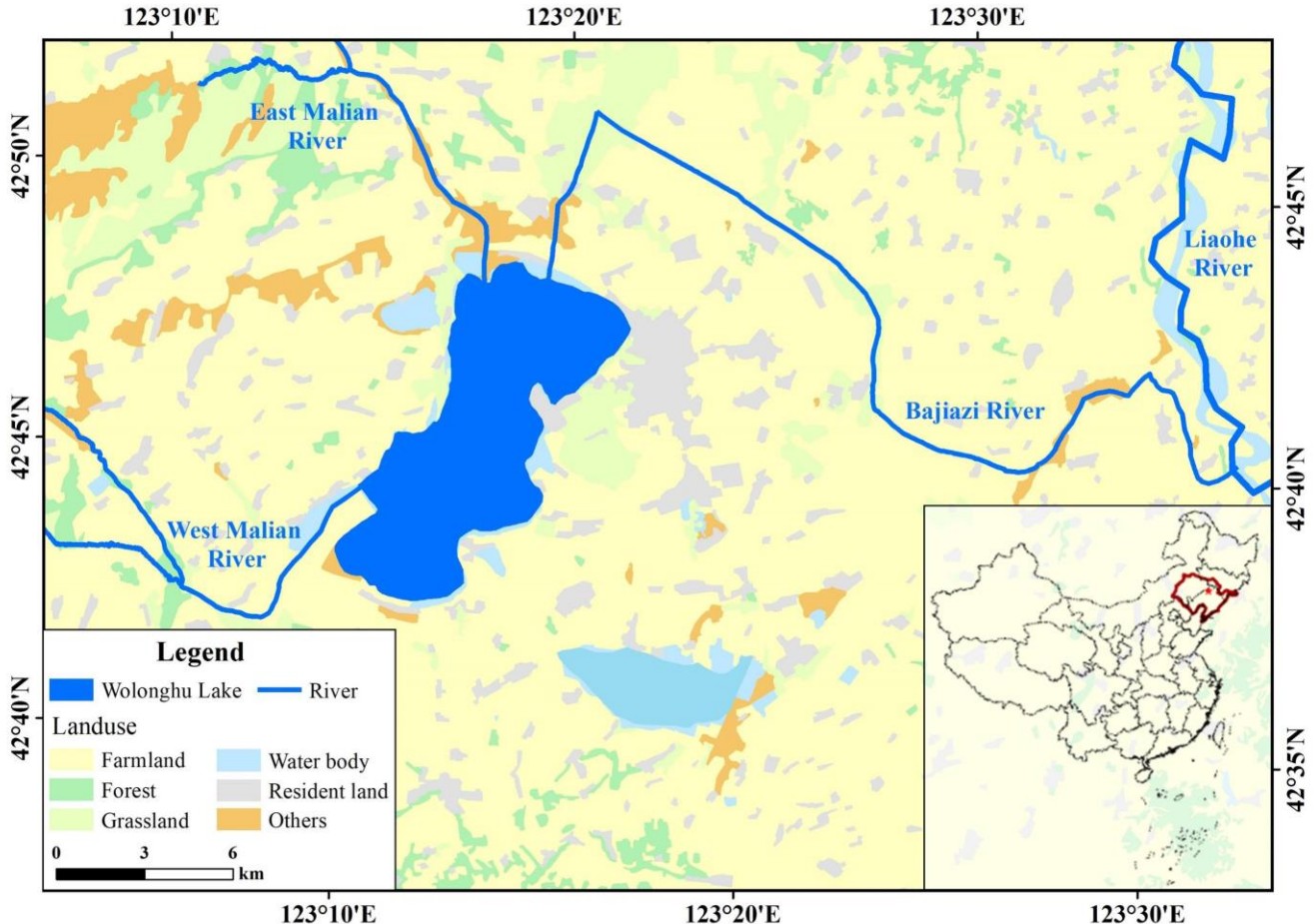

**Figure 3.** The location of the Wolonghu Wetlands.



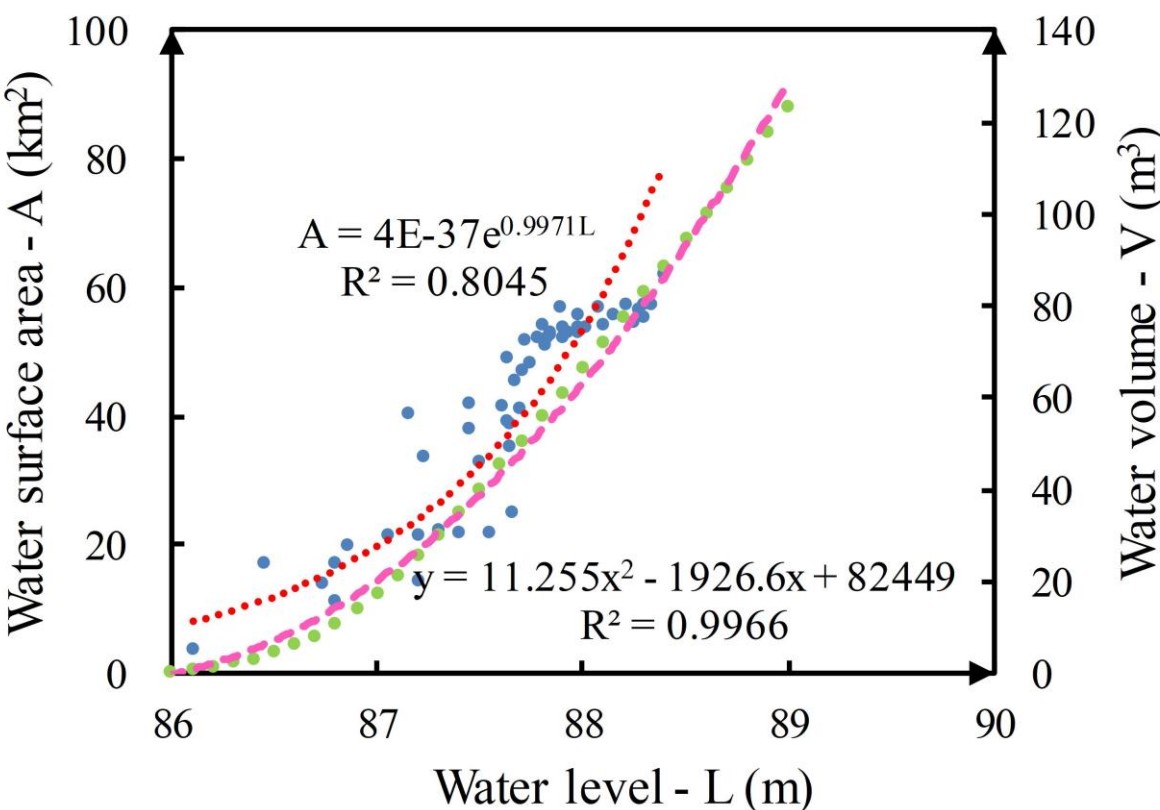

**Figure 4.** The A-L curve and the V-L curve of the Wolonghu Lake. A represents the total water surface area of lake and marsh, V represents the water volume, L represents the water level. The blue points are the observed water surface area with corresponding water level; the red dotted line is the trend line of A-L curve; the green points are the calculated water volume; the pink dashed line is the trend line of V-L curve. The A-L curve and the V-L curve were obtained based on the lake bottom elevation, A and V were calculated with L interval of 0.01 m.





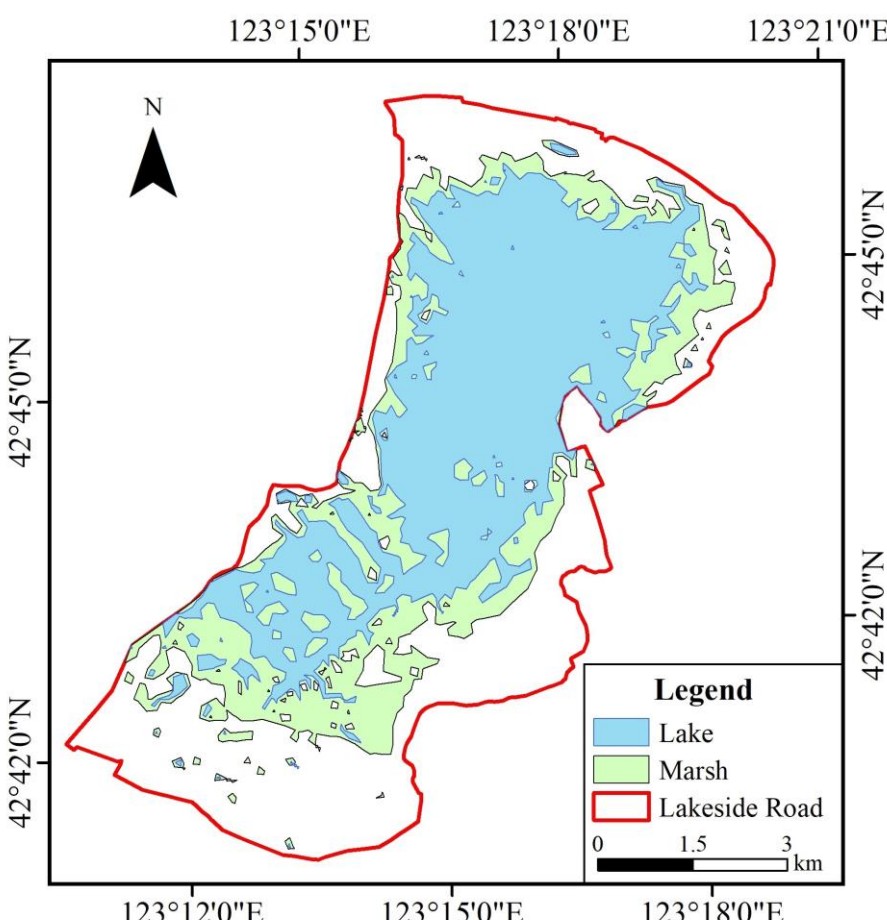

**Figure 5.** The annual optimal lake-marsh pattern of the Wolonghu Wetlands.



**Figure 6.** The monthly optimal lake-marsh pattern of the Wolonghu Wetlands. (a) in January, (b) in February, (c) in Mars, (d) in April, (e) in May, (f) in June, (g) in July, (h) in August, (i) in September, (j) in October, (k) in November, (l) in December.





**Table 1.** Calculation formula of ecological service values (Li et al., 2014; Zheng and Wu, 2015).

| Ecological services | | Calculation formula | Description |
|---|---|---|---|
| Lake | Fishery products | $V_f = \sum Y_{f,i} P_{f,i} Q_{lake}$ | Where $V_f$ is the fishery products value (yuan a⁻¹), $Y_{f,i}$ is the fishery yield of product $i$ (kg m⁻³ a⁻¹), $P_{f,i}$ is the price of product $i$ (yuan kg⁻¹), $Q_{lake}$ is the lake volume (m³) |
| | Pollutants carrying capacity | $V_L = \sum Y_{L,i} P_{L,i} Q_{lake}$ | Where $V_L$ is the pollutants carrying capacity value (yuan a⁻¹), $Y_{L,i}$ is the carrying capacity of pollutant $i$ (kg m⁻³ a⁻¹), $P_{L,i}$ is the reduction price of pollutant $i$ (yuan kg⁻¹), $Q_{lake}$ is the lake volume (m³) |
| | Water conservation | $V_w = P_w \times Q_{lake}$ | Where $V_w$ is the water conservation value (yuan a⁻¹), $P_w$ is the unit storage cost (yuan m⁻³ a⁻¹), $Q_{lake}$ is the lake volume (m³) |
| Marsh | Plant products | $V_p = \sum Y_{p,i} P_{p,i} A_{marsh}$ | Where $V_p$ is the plant products value (yuan km⁻² a⁻¹), $Y_{p,i}$ is the plant yield of product $i$ (kg km⁻² a⁻¹), $P_{p,i}$ is the price of product $i$ (yuan kg⁻¹), $A_{marsh}$ is the marsh area (km²) |
| | Culture | $V_c = P_c \times A_{marsh}$ | Where $V_c$ is the cultural service value (yuan a⁻¹), $P_c$ is the unit price (yuan km⁻² a⁻¹), $A_{marsh}$ is the marsh area (km²) |
| | Carbon sequestration | $V_{CO_2} = \dfrac{1}{2}\left(P_{CO_2} + T_{CO_2}\right) Y_{CO_2} A_{marsh}$ <br> $V_{O_2} = P_{O_2} Y_{O_2} A_{marsh}$ | Where $V_{CO_2}$ is the carbon sequestration value (yuan a⁻¹), $P_{CO_2}$ is the reforestation cost (yuan kg⁻¹), $T_{CO_2}$ is the international tax rate of $CO_2$ (yuan kg⁻¹), $Y_{CO_2}$ is $CO_2$ amount (kg km⁻² a⁻¹), $V_{O_2}$ is the $O_2$ release value (yuan a⁻¹), $P_{O_2}$ is the industrial oxygen production cost (yuan kg⁻¹), $Y_{O_2}$ is the $O_2$ amount (kg km⁻² a⁻¹), $A_{marsh}$ is the marsh area (km²) |
| | Sand fixation | $V_s = \sum Q_{s,i} C_{s,i} A_{marsh}$ | Where $V_s$ is the sand fixation value (yuan km⁻² a⁻¹), $Q_{s,i}$ is sand fixation capacity of vegetation $i$ (t km⁻² a⁻¹), $C_{s,i}$ is the sand removal cost (yuan t⁻¹), $A_{marsh}$ is the marsh area (km²) |
| | Pollution degradation | $V_p = P_p \times A_{marsh}$ | Where $V_p$ is pollution degradation value (yuan a⁻¹), $P_p$ is the unit degradation value (yuan km⁻² a⁻¹), $A_{marsh}$ is the marsh area (km²) |
| | Biodiversity protection | $V_b = P_b \times A_{marsh}$ | Where $V_b$ is the biodiversity protection value (yuan), $P_b$ is the wetland refuge value (yuan km⁻² a⁻¹), $A_{marsh}$ is the marsh area (km²) |

**Table 2.** Data sources.

| Data | Source | Description |
|---|---|---|
| Basic information of Wolonghu Wetlands | Statistical Yearbook of Kanping County | from 2001 to 2016 |
| Meteorology | China Meteorological Data Service Center(http://data.cma.cn) | from 1961 to 2016 |
| Land use | Resource and Environment Data Cloud Platform of the Chinese Academy of Sciences(http://www.resdc.cn) | dataset of 2014 |
| Water resources | Measured and simulated data | from 1961 to 2016 |
| Water quality | Measured data | from 2001 to 2017 |
| Planting and agricultural structure data | Agriculture and Animal Husbandry Bureau of Kangping County | from 2001 to 2016 |



**Table 3.** Minimum water level (min L) and maximum water level (max L) in different month.

| Month | Main protection objects | min L (m) | max L (m) |
|---|---|---|---|
| January | local fish | 87.2 | 88.2 |
| February | local fish | 87.2 | 88.2 |
| March | *Grus leucogeranuss, Scirpus planiculmis Fr. Schmidt* | 87.2 | 87.6 |
| April | *Grus leucogeranuss, Scirpus planiculmis Fr. Schmidt* | 87.2 | 87.6 |
| May | summer migratory birds, emergent plants | 87.2 | 87.8 |
| June | summer migratory birds, emergent plants | 87.2 | 87.8 |
| July | summer migratory birds, emergent plants | 87.2 | 87.8 |
| August | local fish | 87.2 | 88.2 |
| September | local fish | 87.2 | 88.2 |
| October | *Grus leucogeranuss, Scirpus planiculmis Fr. Schmidt* | 87.2 | 87.7 |
| November | *Grus leucogeranuss, Scirpus planiculmis Fr. Schmidt* | 87.2 | 87.7 |
| December | local fish | 87.2 | 88.2 |

**Table 4.** Unit ecological service values of the Wolonghu Wetlands (yuan $m^{-3}$ $a^{-1}$ or yuan $km^{-2}$ $a^{-1}$).

| | Ecological services | Formula description | Value | Subtotal |
|---|---|---|---|---|
| Lake | Fishery products | $Y_{f,i}$ is 0.01 kg $m^{-3}$ $a^{-1}$, $P_{f,i}$ is 30 yuan $kg^{-1}$ | 0.3 | 2.69 |
| | Pollutants carrying capacity | $Y_{L,i}$ is 0.0365 and 0.00547 kg $m^{-3}$ $a^{-1}$ of COD and $NH_3$-N, $P_{L,i}$ is 24 yuan $kg^{-1}$ | 1.01 | |
| | Water conservation | $P_w$ is 1.38 yuan $m^{-3}$ $a^{-1}$ | 1.38 | |
| Marsh | Plant products | $Y_{p,i}$ is 571.34, 16,922.17 and 11,615.57 t $km^{-2}$ $a^{-1}$ of lotus, reed and *Potentilla anserina* L., $P_{p,i}$ is 800 yuan $t^{-1}$ | 23,287,264 | 128,386,864 |
| | Culture | $P_c$ is 5638.4 yuan $km^{-2}$ $a^{-1}$ | 5638.4 | |
| | Carbon sequestration | The average value of $P_{CO_2}$ and $T_{CO_2}$ is 770 yuan $t^{-1}$, $Y_{CO_2}$ is 94,895.6 t $km^{-2}$ $a^{-1}$, $P_{O_2}$ is 0.4 yuan $kg^{-1}$, $Y_{O_2}$ is 69,861.8 t $km^{-2}$ $a^{-1}$ | 101,014,332 | |
| | Sand fixation | $Q_{s,i}$ is 2,120 t $km^{-2}$ $a^{-1}$, $C_{s,i}$ is 170 yuan $t^{-1}$ | 360,400 | |
| | Pollution degradation | $P_p$ is 3,466,910 yuan $km^{-2}$ $a^{-1}$ | 3,466,910 | |
| | Biodiversity protection | $P_b$ is 252,320 yuan $km^{-2}$ $a^{-1}$ | 252,320 | |





**Table 5.** The coefficient value of water demand and water supply in each month in the Wolonghu Wetland.

| t | $\alpha, \beta$ (mm) | $\gamma$ (mm) | $\delta$ (mm) | $Q$ ($10^7$ m³) | $P$ ($10^6$ m³) | $R_S$ ($10^7$ m³) | $R_T$ ($10^6$ m³) | $R_I$ ($10^6$ m³) | $R_G$ ($10^5$ m³) | $V_e$ ($10^7$ m³) |
|---|---|---|---|---|---|---|---|---|---|---|
| January | 20.56 | 0 | 20.56 | 4.845 | 0.135 | 0 | 0.039 | 0.039 | -1.193 | 4.843 |
| February | 35.06 | 0 | 35.06 | 4.857 | 0.184 | 0 | 0.057 | 56976 | -0.477 | 4.843 |
| March | 84.17 | 20.48 | 104.65 | 4.933 | 0.501 | 0 | 0.146 | 0.146 | 3.969 | 4.843 |
| April | 171.56 | 62.75 | 234.32 | 5.303 | 1.339 | 0 | 0.152 | 0.152 | 1.877 | 5.150 |
| May | 240.78 | 106.29 | 347.07 | 5.077 | 2.318 | 0 | 0.981 | 0.981 | 3.268 | 4.813 |
| June | 203.53 | 143.63 | 347.16 | 4.896 | 4.380 | 0 | 2.292 | 2.292 | 4.353 | 4.414 |
| July | 128.84 | 307.80 | 436.65 | 7.347 | 7.443 | 2.291 | 34.575 | 11.661 | 5.011 | 4.261 |
| August | 108.49 | 174.44 | 282.92 | 10.754 | 6.454 | 5.760 | 66.132 | 8.535 | 8.817 | 4.261 |
| September | 122.63 | 112.89 | 235.52 | 5.326 | 2.354 | 0 | 2.621 | 2.621 | 6.332 | 5.027 |
| October | 100.07 | 82.09 | 182.16 | 5.083 | 1.457 | 0 | 0.351 | 0.351 | 0.216 | 4.935 |
| November | 47.81 | 41.06 | 88.88 | 4.879 | 0.514 | 0 | 0.090 | 0.090 | -1.550 | 4.843 |
| December | 24.48 | 0 | 24.48 | 4.851 | 0.190 | 0 | 0.069 | 0.069 | -1.171 | 4.843 |

**Table 6.** Maximum ecological services values of lake-marsh wetland system ($V$) and corresponding area ratio of lake ($\alpha_{lake}$) and marsh ($\alpha_{marsh}$) in different month.

| t | $L$ | $A$ (km²) | $V$ ($10^7$ m³) | $\alpha_{lake}$ | $\alpha_{marsh}$ | $A_{lake}$ (km²) | $A_{marsh}$ (km²) | $V$ ($10^8$ yuan) | $Q$ (m³) |
|---|---|---|---|---|---|---|---|---|---|
| January | 87.61 | 43.656 | 4.755 | 0.738 | 0.262 | 32.236 | 11.420 | 1.328 | 0 |
| February | 87.60 | 43.182 | 4.706 | 0.736 | 0.264 | 31.762 | 11.420 | 1.327 | 15.06 |
| March | 87.57 | 41.736 | 4.553 | 0.658 | 0.342 | 27.481 | 14.255 | 1.627 | 0.007 |
| April | 87.56 | 41.321 | 4.508 | 0.670 | 0.330 | 27.696 | 13.625 | 1.559 | 0.195 |
| May | 87.45 | 37.129 | 4.047 | 0.655 | 0.345 | 24.311 | 12.818 | 1.462 | 0.385 |
| June | 87.43 | 36.394 | 3.964 | 0.634 | 0.366 | 23.069 | 13.325 | 1.515 | 0.018 |
| July | 87.80 | 52.684 | 5.651 | 0.783 | 0.217 | 41.264 | 11.420 | 1.349 | 0 |
| August | 88.20 | 78.585 | 7.823 | 0.855 | 0.145 | 67.165 | 11.420 | 1.397 | 0 |
| September | 87.59 | 42.707 | 4.656 | 0.696 | 0.304 | 29.713 | 12.994 | 1.495 | 0.590 |
| October | 87.57 | 41.862 | 4.566 | 0.714 | 0.286 | 29.893 | 11.969 | 1.383 | 0.215 |
| November | 87.58 | 42.367 | 4.620 | 0.672 | 0.328 | 28.470 | 13.897 | 1.834 | 0.014 |
| December | 87.61 | 43.549 | 4.744 | 0.738 | 0.262 | 32.129 | 11.420 | 1.328 | 1.391 |
| Annual | 87.40 | 35.319 | 3.840 | 0.651 | 0.349 | 23.008 | 12.311 | 16.838 | 0.235 |

