# Peer review of "Determination of the Optimal Lake-Marsh Pattern in the Lake-Marsh Wetland System based on Ecological Land Use and Ecological Water Use"

_Hydrology and Earth System Sciences, 2019_

## Referee Comment (RC1) · Anonymous Referee #1 · 15 Sep 2019

The paper presents an assessment of the combined use of water by a marsh and lake system in China. The authors estimate the best combination of marsh and lake system water use in order to maximise ecological outcomes given a number of restrictions due to the lake operation. While the subject is of potential interest to the readers of HESS, the methods and analysis presented are extremely basic and with very little innovative content. More detailed comments follow. Introduction: the introduction is too long for its content, due to excessive repetition of ideas. There is also insufficient references to optimization methods applied to water resources, which is an important aspect of the manuscript objectives. Methods: The equations presented in the methods are extremely simple and consist essentially of basic water mass balances. These mass

balance equations are combined with ecological service evaluation equations (Table 2) taken from the literature in order to maximise ecosystem services and minimize water shortage. The optimization procedure is not described anywhere, only pointing out to the use of the GAMS software package. No description of optimization method used and criteria for selection. Results: There is very critical analysis of the results, the results section is essentially an enumeration of the model outputs on an annual and monthly basis. There is not much analysis of what would happen if some of the restrictions are relaxed, for example.
* * *

---

## Referee Comment (RC2) · Anonymous Referee #2 · 24 Sep 2019

This study presents a methodology to determine the optimal lake-marsh proportion (not pattern) in a lake-marsh system based on maximising ecosystem services. In my opinion, the paper presents an interesting idea but the implementation is not up to standards for a scientific publication like HESS. There are two main issues of concern: the ecosystem valuation and the optimisation procedure. Both aspects have almost inexistent explanation and there is not enough information to assess the quality or the suitability of the methods used.

The authors used ecosystem valuation estimates taken from Li et al., (2014) and Zheng and Wu, (2015) included in their table 1. The second reference is in Chinese, but the

first one has a collection of data from several lakes in China (not including the study site) not exactly covering the range of services assessed by the authors but still having some similarity. My concern is that Table 1 has about 20 parameters that are site specific, which are later provided in Table 4. The values of the parameters in table 4 are not justified or even compared to the literature (Li et al., 2014 for example shows different values).

The other important aspect that I believe requires more work is the optimization methodology. The exact method is never described or discussed (apparently they used the GAMS software) and Figure 2 is hard to understand without a clear explanation.

Another aspect that have not been included and should be taken into account is that the model assumes that all inundated marsh area has the same ecosystem value, neglecting the fact that not all marsh inundated area will be actually colonised by marsh. Increases in depth of water may generate more inundated marsh areas but it takes time for the colonization to occur.

Last, the paper contains almost no hydrological data. How can the monthly patterns of figure 6 can be analysed without hydrological data? The authors refer to Yan et al (2009) for more hydrological information, which I could not find, but a basic hydrological analysis should have been included in the manuscript
* * *

---

## Author Comment (AC1) · 29 Oct 2019

The authors thank Reviewer 1 for taking the time to review our manuscript, and provide useful feedback. Our responses to each review comment are given **in bold** below.

The paper presents an assessment of the combined use of water by a marsh and lake system in China. The authors estimate the best combination of marsh and lake system water use in order to maximise ecological outcomes given a number of restrictions due to the lake operation. While the subject is of potential interest to the readers of HESS, the methods and analysis presented are extremely basic and with

[Figure]

very little innovative content. More detailed comments follow.

**Response: Thank you for your comments. The methods and analysis seem to be basic, however, our main innovation focuses on the concept and problem-solving approaches. In the context of land use and water resources competitions, how to ensure the minimum water shortage and the maximum ecological services values, is the dilemma we want to solve. It is also a problem that may be encountered in the current ecological protection process. We will improve our manuscript based on your suggestions. Hope to meet with approval.**

Introduction: the introduction is too long for its content, due to excessive repetition of ideas. There is also insufficient references to optimization methods applied to water resources, which is an important aspect of the manuscript objectives.

**Response: Thank you for your suggestion. We agree with the introduction is long as we may emphasize too much on the concept. We will refine the introduction in the revised manuscript. We will follow your suggestion to add relevant references about optimization methods applied to water resources.**

Methods: The equations presented in the methods are extremely simple and consist essentially of basic water mass balances. These mass balance equations are combined with ecological service evaluation equations (Table 2) taken from the literature in order to maximise ecosystem services and minimize water shortage. The optimization procedure is not described anywhere, only pointing out to the use of the GAMS software package. No description of optimization method used and criteria for selection.

**Response: Thank you for your comments. The calculation and optimization procedure in details can be concluded as: First, we calculated the ecological services values and water shortage amount with different area ratio of lake and marsh in a specific water level. The area ratio changed from 0.999:0.001 to 0.001:0.999, with a step of 0.001. The range of water level is from 87.3 m to**

[Figure]

**87.8 m, the calculation step is 0.01 m. Then, under each specific water level condition, we chose the optimal area ratio. The unit ecological services values of marsh is much more greater than lake, also the water shortage amount increases much with increased marsh area. To ensure the total water demand of the system, the optimal lake-marsh pattern considers minimum water shortage amount in priority. Finally, we compared the ecological services values and water shortage amount in all water level conditions, considering the minimum marsh area for wildlife habitat, then chose the optimal pattern. The details will be added in the revised manuscript.**

Results: There is very critical analysis of the results, the results section is essentially an enumeration of the model outputs on an annual and monthly basis. There is not much analysis of what would happen if some of the restrictions are relaxed, for example.

**Response: Thank you for your advice. We will follow your advice to analyze what would happen and propose some allocation plans if some of the restrictions are relaxed.**

---

## Author Comment (AC2) · 29 Oct 2019

The authors thank Reviewer 2 for taking the time to review our manuscript, and provide useful feedback. Our responses to each review comment are given **in bold** below.

This study presents a methodology to determine the optimal lake-marsh proportion (not pattern) in a lake-marsh system based on maximising ecosystem services. In my opinion, the paper presents an interesting idea but the implementation is not up to standards for a scientific publication like HESS. There are two main issues of concern: the ecosystem valuation and the optimisation procedure. Both aspects have almost

[Figure]

inexistent explanation and there is not enough information to assess the quality or the suitability of the methods used.

**Response: Thank you for your comments. We will explain more about the ecosystem valuation and the optimization procedure in the revised manuscript. We will follow your suggestion to add more information to assess the quality and the suitability of the methods used.**

The authors used ecosystem valuation estimates taken from Li et al., (2014) and Zheng and Wu, (2015) included in their table 1. The second reference is in Chinese, but the first one has a collection of data from several lakes in China (not including the study site) not exactly covering the range of services assessed by the authors but still having some similarity. My concern is that Table 1 has about 20 parameters that are site specific, which are later provided in Table 4. The values of the parameters in table 4 are not justified or even compared to the literature (Li et al., 2014 for example shows different values).

**Response: Thank you for your comments. We tried to find references in English about the evaluation of the ecological services values in the Wolonghu Wetlands, but there are none. Thus, we added the references of Li et al. (2014) and Zheng and Wu (2015). We proposed our evaluation method based on both references. As for the value of the site specific 20 parameters, some can be found in Zheng and Wu (2015), others were obtained by our field investigation (we will point out these values in the revised version). We will follow your suggestion to justify and compared to previous literatures.**

The other important aspect that I believe requires more work is the optimization methodology. The exact method is never described or discussed (apparently they used the GAMS software) and Figure 2 is hard to understand without a clear explanation.

**Response: Thank you for your comments. The calculation and optimization**

[Figure]

**procedure in details can be concluded as: First, we calculated the ecological services values and water shortage amount with different area ratio of lake and marsh in a specific water level. The area ratio changed from 0.999:0.001 to 0.001:0.999, with a step of 0.001. The range of water level is from 87.3 m to 87.8 m, the calculation step is 0.01 m. Then, under each specific water level condition, we chose the optimal area ratio. The unit ecological services values of marsh is much more greater than lake, also the water shortage amount increases much with increased marsh area. To ensure the total water demand of the system, the optimal lake-marsh pattern considers minimum water shortage amount in priority. Finally, we compared the ecological services values and water shortage amount in all water level conditions, considering the minimum marsh area for wildlife habitat, then chose the optimal pattern. The details will be added in the revised manuscript.**

**For Figure 2, we added much explanation: "The specific calculation process contains six parts: i) determine the protection objects; ii) analyze the required water level of each protection object; iii) determine min L and max L; iv) calculate unit ecological services values of lake and marsh; v) calculate the water demand amount and water supply amount; vi) obtain the optimal area ratio of lake and marsh, also the corresponding water level. Notably, the total water surface area of lake and marsh is definite for a certain water level."**

Another aspect that have not been included and should be taken into account is that the model assumes that all inundated marsh area has the same ecosystem value, neglecting the fact that not all marsh inundated area will be actually colonised by marsh. Increases in depth of water may generate more inundated marsh areas but it takes time for the colonization to occur.

**Response: Thank you for your throughout comments. We agree with your opinion. The calculation of ecological services values was simplified, we didn't consider too much about plant growing time. We tried to solve this issue but**
**it's complex. Also, there are scant previous studies solving this question. The unit ecological services value of marsh considers the average value, which may reflect the overall level.**

Last, the paper contains almost no hydrological data. How can the monthly patterns of figure 6 can be analysed without hydrological data? The authors refer to Yan et al (2009) for more hydrological information, which I could not find, but a basic hydrological analysis should have been included in the manuscript.

**Response: Thank you for your question. We are sorry for making mistake about the publication time of the reference of Yan et al. The sentence will be changed as "The analysis of water balance in the Wolonghu Lake can be found in the reference of Yan et al. (2019)." The study of Yan et el. (2019) was accepted in January, 2019 while published in June, 2019 (after our HESSD paper published), the final paper title has changed to "Water balance in the Wolonghu Lake". We will add relevant hydrological data in the revised manuscript. Hope to meet with approval.**